# Rare Sugar Metabolism and Impact on Insulin Sensitivity along the Gut–Liver–Muscle Axis In Vitro

**DOI:** 10.3390/nu15071593

**Published:** 2023-03-25

**Authors:** Amar van Laar, Charlotte Grootaert, Andreja Rajkovic, Tom Desmet, Koen Beerens, John Van Camp

**Affiliations:** 1NutriFOODChem, Department of Food Technology, Safety and Health, Faculty of Bioscience Engineering, Ghent University, 9000 Ghent, Belgium; 2Food Microbiology and Food Preservation, Department of Food Technology, Safety and Health, Faculty of Bioscience Engineering, Ghent University, 9000 Ghent, Belgium; 3Centre for Synthetic Biology, Department of Biotechnology, Faculty of Bioscience Engineering, Ghent University, 9000 Ghent, Belgium

**Keywords:** rare sugars, diabetes, skeletal muscle, bioenergetics, cell research

## Abstract

Rare sugars have recently attracted attention as potential sugar replacers. Understanding the biochemical and biological behavior of these sugars is of importance in (novel) food formulations and prevention of type 2 diabetes. In this study, we investigated whether rare sugars may positively affect intestinal and liver metabolism, as well as muscle insulin sensitivity, compared to conventional sugars. Rare disaccharide digestibility, hepatic metabolism of monosaccharides (respirometry) and the effects of sugars on skeletal muscle insulin sensitivity (impaired glucose uptake) were investigated in, respectively, Caco-2, HepG2 and L6 cells or a triple coculture model with these cells. Glucose and fructose, but not l-arabinose, acutely increased extracellular acidification rate (ECAR) responses in HepG2 cells and impaired glucose uptake in L6 cells following a 24 h exposure at 28 mM. Cellular bioenergetics and digestion experiments with Caco-2 cells indicate that especially trehalose (α1-1α), D-Glc-α1,2-D-Gal, D-Glc-α1,2-D-Rib and D-Glc-α1,3-L-Ara experience delayed digestion and reduced cellular impact compared to maltose (α1-4), without differences on insulin-stimulated glucose uptake in a short-term setup with a Caco-2/HepG2/L6 triple coculture. These results suggest a potential for l-arabinose and specific rare disaccharides to improve metabolic health; however, additional in vivo research with longer sugar exposures should confirm their beneficial impact on insulin sensitivity in humans.

## 1. Introduction

Diabetes is a metabolic disease that was estimated to affect more than 500 million in 2021, with a further increase projected for the future [1]. In type 2 diabetes, as the most common and lifestyle-dependent variant, the cellular actions of insulin including its stimulatory effect on glucose uptake is impaired, resulting in increased blood glucose levels (hyperglycemia) [2,3,4]. Metabolic abnormalities in diabetes may induce oxidative stress, inflammation and production of advanced glycation end products (AGEs), which all contribute to vascular damage and thereby increase the risk for cardiovascular diseases, kidney failure and blindness [5,6]. Before diabetes is diagnosed, peripheral insulin sensitivity is often already impacted, and lifestyle-related prevention is especially effective in this prediabetic state [7]. Furthermore, prediabetes already increases the risk for vascular complications [8]. In this context, sugar replacement is of interest, as (I) excessive sugar intake may contribute to obesity as an important risk factor for type 2 diabetes, (II) low-glycemic index diets may decrease the risk for type 2 diabetes development, (III) a lower intake of glycemic sugars could limit the high blood glucose people in people with impaired insulin sensitivity and (IV) a mechanistic basis exists by which high amounts of fructose impact hepatic de novo lipogenesis and the development of hepatic insulin resistance [9,10,11,12]. Relatively new candidates to substitute conventional sugars can be found among rare sugars, which have been defined by the International Society of Rare Sugars as ‘monosaccharides and their derivatives that are present in limited quantities in nature’ [13]. Several studies have reported benefits of rare compared to conventional sugars such as a slower digestion and reduced impact of isomaltulose on blood glucose levels compared to sucrose [14].

The regulation of metabolic health to prevent metabolic abnormalities involves an intensive interaction between the liver and skeletal muscles, which depends on available metabolic substrates and inter-organ crosstalk [15,16]. Liver health impacts the ability of skeletal muscles to maintain glucose homeostasis, which is crucial as (resting) skeletal muscles are responsible for 75% of insulin-mediated whole-body glucose disposal and regulate many metabolic health factors via secreted myokines [17,18]. Skeletal muscle insulin resistance impairs this insulin-mediated glucose uptake and thereby increases the glycemic effect of foods which for sugars depends partially on intestinal digestion speed and resulting monosaccharide uptake, and partially on how fast glucose is cleared from the circulation [19]. Additionally, glucose clearance in skeletal muscles is impacted by exercise, which promotes insulin-independent uptake of glucose and temporarily improves insulin sensitivity [20].

Insulin sensitivity is closely related to mitochondrial respiration. Mitochondrial dysfunction may contribute to insulin resistance due to inflammation, ROS production and β-oxidation impairment, whereas treatments to improve mitochondrial function also alleviate insulin resistance [21]. Furthermore, insulin as such improves the mitochondrial function, which is prevented by palmitate as a known inducer of insulin resistance [22]. Insulin resistance is often characterized by a reduced metabolic flexibility as well, which is the capacity to adapt fuel oxidation depending on the available substrates, such as glucose and fatty acids; therefore, decreased metabolic flexibility is recognized as a predictor for metabolic disease development [23]. Therefore, cellular bioenergetics, which is the study of pathways via which cells generate and invest energy [24], is an interesting approach to investigate insulin sensitivity, along with insulin-mediated glucose uptake. Cellular bioenergetics can be studied with respirometry, using probes that measure oxygen and protons [24], which result in an extracellular acidification rate (ECAR) as an indirect measure of anaerobic lactate production and oxygen consumption rate (OCR) as an indirect measure of mitochondrial respiration. Metabolic stressors are often used to obtain further insight in metabolic pathways by eliminating the metabolism of specific macronutrients or by inhibiting the electron transport chain, as visualized in Figure 1. Hereby, respirometry has revealed remarkable metabolic adaptations upon the replacement of glucose as the dominant sugar in cell culture. For instance, chronic replacement of glucose by galactose skewed muscle cells towards a more aerobic metabolism, a transition that was exclusively observed in metabolically healthy myotubes and absent in post-diabetic myotubes [25]. However, knowledge on cellular adaptations upon chronic replacement of glucose by other sugars is scarce.

In this study, we aim to investigate the impact of rare and conventional sugars on metabolic health by studying (I) skeletal muscle insulin sensitivity, (II) acute sugar-induced changes in cellular bioenergetics and (III) metabolic adaptation upon chronic exposure of liver cells to particular monosaccharides.

## 2. Materials and Methods

### 2.1. Materials

Caco-2, HepG2 and L6 cell lines were obtained from ATCC (Manassas, VA, USA). Dulbecco’s Modified Eagle Medium (DMEM), Phosphate Buffered Saline (PBS), non-essential amino acids, trypsin-EDTA and Penicillin/Streptomycin were purchased from Gibco (Paisley, UK). XF base medium (with and without (4-(2-hydroxyethyl)-1-piperazineethanesulfonic acid (HEPES)), Seahorse plates (24-well), Seahorse cartridges (24-well), Seahorse calibrant and mito-stress assay kits were all obtained from Agilent (Machelen, Belgium). Fetal Bovine Serum (FBS) and Trypan Blue were obtained from VWR (Leuven, Belgium). Cell culture flasks (25 and 75 cm^2^) and some cell culture plates (96-well, 6-well and 6-well coculture ThinCerts) were obtained from Greiner (Vilvoorde, Belgium). Additionally, 24-well plates and Transwell plates with 24-well insert were obtained from Novolab (Diebeke, Belgium), whereas 12-well coculture inserts were purchased from Sigma-Aldrich (Overijse, Belgium). Human recombinant insulin, 2-deoxyglucose, glycerol, 2-mercaptoethanol, bovine serum albumin (BSA), glucose, fructose, maltose (D-Glcp-α1,4-D-Glcp, 99% pure), galactose, mannitol, resazurin, trishydroxymethylaminomethane (Tris), sulforhodamine B (SRB), horseradish peroxidase and glucose oxidase from Aspergillus niger were also purchased from Sigma-Aldrich. Rare disaccharides nigerose (D-Glcp-α1,3-D-Glcp, 88% pure), kojibiose (D-Glcp-α1,2-D-Glcp, 99% pure), D-Glcp-α1,2-D-Galp (96% pure), D-Glcp-α1,2-D-Ribp (95% pure) and D-Glcp-α1,3-L-Arap (95% pure) were synthesized as described previously [26,27], whereas trehalose (D-Glcp-α1,1α-D-Glcp, 99% pure) was kindly provided by Cargill (Mechelen, Belgium). l-arabinose was obtained from Merck (Darmstadt, Germany). NaHCO3 and glacial acetic acid and sodium acetate were obtained from Chem-Lab (Zedelgem, Belgium). Dodecyl sulfate (SDS), 2,2′-azino-bis(3-ethylbenzothiazoline-6-sulfonic acid (ABTS) and OnGuard II Ag Crtgs were purchased from ThermoFisher (Merelbeke, Belgium). Trichloroacetic acid (TCA) was purchased from Acros Organics (Geel, Belgium). DC protein assay kit was purchased from Bio-Rad (Temse, Belgium).

### 2.2. Cell Culture and Exposure

Caco-2, HepG2 and L6 cells were cultured in DMEM with 25 mM (Caco-2 cells) or 5.5 mM glucose (HepG2 and L6 cells), supplemented with 10% heat-inactivated and sterile-filtered FBS, 1% non-essential amino acids and 1% Penicillin/Streptomycin. Furthermore, alternative culture conditions were tested for HepG2 cells (5.5 mM galactose, 5.5 mM fructose, 5.5 mM l-arabinose or 25 mM glucose as sole carbohydrate source), which were applied for 2–3 weeks prior to cell seeding. Cells were incubated in a CO_2_ incubator (Memmert; VWR, Belgium) at 37 °C and 10% CO_2_, and the cell medium was refreshed every two or three days. At 80% confluency, cells were trypsinized and split at a ratio of 1:3 (Caco-2), 1:5 (HepG2) or 1:10 (L6 cells). The cell suspension was mixed 1:1 with Trypan Blue and cells were counted using a Bürker counting chamber (VWR; Leuven, Belgium).

Monocultures of HepG2, L6 and Caco-2 cells were seeded at a density of 3 × 10^4^ cells per well in 24-well Seahorse plates or 2 × 10^4^ cells per well in 96-well plates. HepG2 and L6 cells were cocultured in a 24-well coculture setup, in which HepG2 cells were seeded on the transwell insert and L6 cells in the basolateral compartment, both at a density of 5 × 10^4^/well. Finally, triple cocultures with Caco-2, HepG2 and L6 cells were established (Figure 2). In this setup, Caco-2 cells were seeded in the upper 12-well inserts at 1.5 × 10^5^ cells per well, HepG2 cells were seeded in the middle 6-well inserts at 3 × 10^5^ cells per well, and L6 cells on the bottom of the setup were seeded at 3 × 10^5^ cells per well in normal 6-well cell culture plates. Cell layers were divided by two metal supports holding, respectively, the Caco-2 and HepG2 inserts. HepG2 and L6 cells were used upon confluency, whereas Caco-2 cells were used after two weeks of differentiation. The cells were exposed in sugar-free XF base medium with 3.7 g/L NaHCO3 and sugars of interest at concentrations up to 28 mM for monosaccharides or 14 mM as the disaccharide equivalent, as the only supplements.

### 2.3. Development of Insulin Resistance Models

Cellular insulin sensitivity was investigated in multiple setups using various methods, as visualized in Figure 2.

As a first experiment, the effect of insulin on glucose uptake was tested for a range of insulin concentrations (1.67–1.67 × 10^4^ μIU/mL). In a next step, the effect of 24 h exposure to physiological (5.5 mM) versus high glucose (25 mM) was tested in a L6 96-well monoculture model. These are the glucose concentrations that are typically found in commercial DMEM cell culture media. This monoculture model was expanded by adding HepG2 liver cells in a 24-well coculture setup. HepG2 cells were exposed for 24 h to monosaccharides at 28 mM, allowing an indirect exposure of metabolized monosaccharides on the L6 muscle cells. Finally, Caco-2 cells were added to the model, resulting in a 6-well triple-coculture setup with Caco-2, HepG2 and L6 cells. In this model, Caco-2 cells were exposed for 24 h to different disaccharides at 14 mM, and their impact on insulin-stimulated glucose uptake and insulin pathway signaling was investigated in L6 muscle cells. Following exposures in different models, an additional 24 h exposure to 25 mM glucose in combination with 1.67 or 50 μIU/mL insulin was performed to determine insulin-mediated glucose uptake from measurements of glucose concentration in the cell-treated medium, as a measure for insulin sensitivity in L6 cells. In addition, alterations of insulin-mediated aerobic and anaerobic responses following 24 h sugar exposure were investigated using respirometry with a Seahorse XFe24 analyzer.

### 2.4. HPAEC-PAD Sugar Analysis to Study Digestibility of Rare Disaccharides

Medium samples of 20 μL were taken from the Caco-2 compartment of the triple coculture model after 0, 2, 6 and 24 h of 14 mM disaccharide exposure. These samples were diluted 1000 times in distilled water and 1 mL of diluted sample was filtered through OnGuard II Ag Crtgs to remove negatively charged ions. Filtered samples were used to determine glucose concentrations in the samples using high-pressure anion-exchange chromatography with pulsed amperometric detection (HPAEC-PAD) [28]. Sample analysis was performed using a HPAEC-PAD system (Dionex ICS-3000, Thermo Scientific, (Merelbeke, Belgium)) with a CarboPac PA20 pH-stable anion exchange column for carbohydrate separation at a flow rate of 0.5 mL·min^−1^. An isocratic elution with 99% eluent B (100 mM NaOH) and 1% eluent C (1 M NaOAc and 100 mM NaOH) was used for the first 9 min, followed by 70% eluent B and 30% eluent C for 3 min. The initial eluent composition was then restored to run samples for 1.5 min. Glucose concentrations were quantified using a standard curve (1–30 μM) with an LOD of 0.14 μM and LOQ of 0.43 μM.

### 2.5. Aerobic and Anaerobic Metabolism

Respirometry experiments were performed with different setups and stressors as visualized in Figure 3a. As visualized, respirometry was used for three different end points: to characterize (I) the speed and potency of the effects of different (rare) disaccharides on anaerobic responses in intestinal Caco-2 cells, as an indirect measure of brush border digestion; (II) origin of ATP production (aerobic and/or anaerobic) upon short-term incubation with different monosaccharides in HepG2 liver cells; and (III) insulin-stimulated anaerobic glycolysis and maximal respiration in L6 cells within the context of skeletal muscle insulin resistance. Examples of ECAR and OCR profiles with these setups, based on our experimental data under standard conditions, are shown in Figure 3b.

#### 2.5.1. General Seahorse Setup

In all setups, Seahorse cartridges were hydrated with 1 mL/well calibrant solution and placed in a closed plastic bag with wet paper to prevent dehydration during a 24 h incubation in an incubator at 37 °C. On the day of the assay, calibrant solution was refreshed 1 h before loading cartridges, and the injection ports of the cartridges were loaded with treatment solutions at ten times the final concentration, with increasing volumes for each port (A 20 μL, B 22 μL, C 25 μL and D 28 μL). Cells were placed on nutrient-free (no glucose, glutamine or pyruvate) base XF medium with HEPES pH7.4 for one hour under CO_2_-free conditions at 37 °C. Immediately before the assay, cells were washed again and received 180 μL sugar-free XF medium with HEPES. The Seahorse procedure was run with fixed durations for all assays: three to five loops (fiver loops were only used for one of the disaccharide digestion experiments) of mixing (±1 min), waiting (±2 min) and measuring (±5 min) after each injection.

#### 2.5.2. Origin of ATP Production: Setup and Calculations

The ATP rate assay (Agilent) was used to determine the amount and origin of ATP production during exposures with glucose, fructose, galactose, l-arabinose and mannitol in HepG2 cells, according to the manufacturer’s instructions. To investigate the impact of monosaccharides on the ATP rate in standard HepG2 cells and cells that were chronically exposed to alternative monosaccharides, they were injected at 10 mM. Next, the response on oligomycin (1.5 µM) and rotenone/antimycin A (0.5 µM) was measured. ATP rates were calculated according to the assay manual [29] and Seahorse white papers [30], and are based on changes within three readings (from the moment of injection until the final measurement before the next injection is applied) after injection of sugars or stressors. The mitochondrial energy production rate (ATPmito) is calculated as the component of oxidative phosphorylation that is inhibited by introduction of oligomycin (OCR energy, Figure 3 and Equation (1)), multiplied by the number of oxygen molecules (2) and the phosphate per oxygen ratio (P/O is 2.75 on average, Equation (2)). Calculation of the glycolytic ATP production rate requires the total proton efflux rate (PERtotal), which is obtained by multiplying ECAR with the volume of the measurement chamber (5.65 μL in 24-well format), volume scaling factor (1.19 μL in 24-well format) and the buffer factor (depends on medium composition and sensor, can be determined manually, and should be between 2.6–4, Equation (3)). OCRmito is then calculated as the part of OCR that is inhibited after the introduction of both oligomycin and rotenone/antimycin A (Figure 3 and Equation (4)). PERmito is the CO_2_-dependent non-glycolytic proton efflux and can be calculated by multiplying the mitochondrial OCR (OCRmito) with the CO_2_ conversion factor, which Agilent determined to be 0.6 for the 24-well system (Equation (5)). The glycolytic proton efflux rate (PERglyco) is calculated by subtracting the mitochondrial proton efflux (PERmito) from the total proton efflux rate (PERtotal) (Figure 3 and Equation (6)), and is the equivalent to the glycolytic proton efflux rate (PERglyco) (Equation (7)), in case the medium and stressors indicated in the ATP rate manual protocol are used. Finally, the ATP rate (ATPtotal) was calculated as the sum of the mitochondrial (ATPmito) and glycolytic ATP production rate (ATPglyco) (Equation (8)).
OCRenergy = OCRbasal − OCRoligo(1)
ATPmito = OCRenergy × 2 × 2.75 (5.5 × OCRenergy)(2)
PERtotal = ECAR × 1.19 × 5.65 × 2.8 (18.8 × ECAR)(3)
OCRmito = OCRbasal − OCRrot/antA(4)
PERmito = OCRmito × 0.6(5)
PERglyco = PERtotal − PERmito(6)
ATPglyco = PERglyco(7)
ATPtotal = ATPmito + ATPglyco(8)

#### 2.5.3. Disaccharide Digestion

Caco-2 cells were exposed to disaccharides twice (15 or 25 min exposure each) within the Seahorse running time, after which the anaerobic responses were stopped with 50 mM 2-deoxyglucose (Figure 3a). Cellular respirometry is usually performed with semi-confluent cells, but we determined ECAR responses in a confluent layer of differentiated Caco-2 cells to induce additional α-glucosidase expression.

#### 2.5.4. Insulin-Mediated Responses and Insulin Sensitivity

L6 cells were exposed to insulin for 20 min at 50–1.67 × 10^4^ μIU/mL. Then, a variant of the mito-stress assay was performed with glucose as first injection, followed by oligomycin (1.5 μM), FCCP (0.5 μM) and rotenone/antimycinA (0.5 μM) (Figure 3a). In experiments to test the effect of high glucose pre-treatment on insulin-mediated responses, an additional 24-h glucose exposure at 28 mM was performed prior to the insulin exposure.

### 2.6. Insulin-Mediated Glucose Uptake Based on GOD-POD Measurements

A glucose oxidase-peroxidase (GOD-POD) mixture [31] was prepared by adding 50 mg ABTS, 45.23 mg glucose oxidase and 6.92 mg peroxidase in 100 mL 0.2 M acetic acid (pH 4.5), which was stored in aliquots at −20 °C. The 50 μL undiluted cell-treated medium from L6 cells was added to a clear plate, after which 200 μL GOD-POD reagent mixture was added. The plate was incubated for 5 min at 37 °C and absorbance was measured at 420 nm. A glucose standard curve (0–2400 μM in distilled water) was prepared to quantify the amount of glucose in cell-treated medium with a LOD and LOQ of 53 μM and 160 μM, respectively. Glucose uptake was calculated for each exposure condition by subtracting the calculated glucose content in cell-treated medium from the calculated glucose content in medium from wells without cells as a blank.

### 2.7. Resazurin Assay for Cellular Reductase Activity

Resazurin stock solution (1 mg/mL in distilled water) was added to the cell medium at 1:100 *v*/*v* [32]. The plate was incubated for two hours at 37 °C and fluorescence was measured (λexc/λem = 560/590 nm) in a black 96-well plate.

### 2.8. Protein Correction

Results were corrected for protein content by performing an SRB assay [33] or BioRad protein assay [34].

#### 2.8.1. SRB Assay

After the assays, cells were fixated with 1:4 *v*/*v* 50% TCA in medium for at least 1 h at 4 °C. The cells were washed with tap water at least three times and SRB solution was added in excess. After 30 min, the plate was washed at least three times with 1% glacial acetic acid. Next, the protein-adhered SRB stain was dissolved by adding 200 μL 10 mM Tris and pipetted up and down to homogenize the stain. Absorbance was measured at 490 nm.

#### 2.8.2. Lysate Preparation and Bio-Rad Protein Assay

After 24 h exposure, L6 cells were first washed with cold PBS and 600 μL Laemmli buffer (1.5×) was added per well. Then, the cell layer was disrupted with cell scraper and the lysate was transferred to an Eppendorf tube and centrifuged for 10 min at 14,000 rpm and 4 °C. The supernatant was transferred to a second Eppendorf tube on ice, and stored at −20 °C prior to analysis. Protein content of the lysates was determined with the Bio-Rad DC protein assay according to the manufacturer’s instructions. Standard curves (0.2–3 mg/mL) were constructed from bovine serum albumin (BSA) standard solution, showing a linear relationship between absorbance and protein content with a LOD and LOQ of 0.09 and 0.27 mg/mL, respectively. Absorbance at 750 nm was measured after 15 min incubation of the lysate with the reaction mixture.

### 2.9. Statistics and Calculations

Statistical analyses were performed with SPSS 26 using a significance cut-off of *p* < 0.05. Levene’s tests were performed to check for homogeneity of variance. Conditions were compared with one-way analysis of variance (ANOVA), using the Tukey correction for homogeneous data or Games–Howell correction for non-homogenous data. As an exception, the significance of changes in glucose concentrations (HPAEC-PAD data) was determined within exposure conditions using two-way ANOVA, with time and concentration as input parameters. Mitochondrial, glycolytic and total ATP rates were calculated as explained stepwise in the paragraph on ‘aerobic and anaerobic metabolism’. Insulin concentrations were converted from pM to μIU/mL using a conversion factor of 6, based on the molecular weight of 5808 kDa.

## 3. Results

### 3.1. Disaccharide Digestion and Related ECAR Responses in Intestinal Caco-2 Cells

Respirometry was used to study how gradual glucose release from easily and slowly digestible disaccharides impacts ECAR responses. In differentiated Caco-2 cells, injection of 10 mM glucose and to a lesser degree 5 mM maltose resulted in an increase in ECAR, whereas no response was observed with 10 mM of the mannitol control or kojibiose and trehalose at 5 mM (Figure 4). A second injection at three times the initial concentration did not have clear additive effects. Sugar-induced ECAR responses were stopped upon injection of 50 mM deoxyglucose. The absence of an increase in ECAR with kojibiose and trehalose suggests that these rare sugars have a reduced metabolic impact compared to glucose, most likely related to delayed digestion.

HPAEC-PAD glucose quantification showed differences in glucose release for different rare disaccharides. Maltose digestion resulted in a significant increase in glucose concentrations in the Caco-2 medium, which was not observed for the mannitol or rare disaccharides (Figure 5). This pronounced increase in medium glucose concentrations during maltose exposure suggests that the release of glucose exceeds cellular uptake, highlighting that this sugar is more easily digested than the rare disaccharides. Kojibiose and nigerose digestion resulted in stable glucose concentrations over time, whereas a decrease in glucose concentrations was observed during 24-h exposure to trehalose and the analogues of kojibiose (D-Glc-α1,2-D-Gal and D-Glc-α1,2-D-Rib) and nigerose (D-Glc-α1,3-L-Ara). The stable glucose concentrations during kojibiose and nigerose exposure suggests that these sugars are digested at an intermediate rate, whereas a decrease in glucose concentrations suggests slow digestion of a sugar, insufficient to supply the cells with basal levels of glucose.

### 3.2. Effects of Chronic Monosaccharide Exposure on Energy Metabolism in HepG2 Liver Cells

#### 3.2.1. Cell Growth and Morphology

To study which aspects of the cellular metabolism are impacted by chronic exposure to only glucose, galactose, fructose and l-arabinose, chronically exposed HepG2 cells were characterized in terms of their basal state (in the absence of nutrients) and after additional exposure to the different monosaccharides (pre-treatment + extra 24-h exposure to different monosaccharides) using resazurin conversion and respirometry. Replacement of glucose by other monosaccharides reduced the growth rate of HepG2 cells, with the largest decrease following chronic l-arabinose exposure (at least four-fold slower, based on cell count and time till ±80% confluency), and an intermediate cell growth (two-to-three-fold slower) following chronic exposure to galactose and fructose. Two days after splitting, cells that were chronically exposed to fructose, galactose or l-arabinose were in a different stage of growth with a different morphology (Figure 6), although they eventually obtained ‘normal’ HepG2 morphology as confluency increased. These findings suggest that cells grow most efficiently in a glucose-containing medium, but are able to adapt to the presence of other sugars.

#### 3.2.2. Resazurin Conversion

Chronic exposure to alternative monosaccharides at 5.5 mM altered protein-corrected resazurin conversion in HepG2 cells as well, although the same monosaccharides at 28 mM (glucose and fructose) provided stimulatory 24 h energy effects (compared to mannitol) in all chronically pre-treated HepG2 cells (Table 1). Galactose approached the fructose and glucose response specifically in cells that were chronically pre-treated with galactose, suggesting an adaptation to the presence of galactose. Galactose pre-treated cells also produced more energy than the traditionally cultured HepG2 cells (with 5.5 mM glucose) during all of the 24 h exposures (including mannitol) (Table 1), indicating an alteration of the basal cellular metabolism. These increased responses were also observed in l-arabinose pre-treated cells, except during exposure to l-arabinose. Fructose pre-treated HepG2 cells responded more strongly to specifically a glucose exposure compared to traditionally cultured cells.

#### 3.2.3. Acute OCR and ECAR Responses to Sugar Injections in Galactose Versus Glucose Pre-Treated Cells

To study differences in cellular metabolism and contribution of glycolysis and/or mitochondrial respiration on in vitro ATP production of structurally different monosaccharides, more specific glucose, fructose, galactose and l-arabinose, and ATP rates were determined with respirometry. In HepG2 cells long-term pre-treated with 5.5 mM glucose or galactose, injection of 10 mM glucose or fructose resulted in an acute and direct increase in ECAR and glycolytic ATP production, which was not observed upon mannitol, galactose or l-arabinose injection (Figure 7a,b,e,f). Injection of glucose, but not fructose, also resulted in a significant increase in the total ATP production rate (Figure 7e,f). Galactose and l-arabinose did not alter any of the ATP rates in either glucose or galactose pre-treated HepG2 cells, suggesting that these sugars have little impact on the liver metabolism. Oligomycin further increased anaerobic glycolysis slightly in cells exposed to glucose, but had a significantly different impact on cells exposed to fructose, in which it reduced anaerobic glycolysis (Figure 7a). HepG2 cells pre-treated with galactose had higher basal OCR and lower ECAR levels before injection of the sugars, suggesting that these cells are in a more aerobic state. Upon injection of glucose, these cells also provided more potent ECAR responses and experienced a more pronounced decrease in OCR (Figure 7).

### 3.3. Insulin Sensitivity and Glucose Uptake in L6 Muscle Cells

To investigate how sugars impact insulin sensitivity in skeletal muscle cells, insulin-stimulated glucose uptake was determined, starting with a L6 monoculture and gradually building a Caco-2/HepG2/L6 triple coculture model.

#### 3.3.1. Insulin-Mediated Glucose Uptake Determined with the GOD-POD Assay

Insulin (24 h exposure) increased glucose uptake in L6 muscle cells, starting from concentrations of 1.67 μIU/mL (Figure 8a). In an experiment to test the impact of a hyperglycemic environment on insulin sensitivity in L6 cells, exposure to 25 mM glucose for 24 h significantly reduced (±23%) insulin-stimulated glucose uptake, compared to cells exposed to 5.5 mM glucose (Figure 8b). In a HepG2/L6 coculture using the same exposure regimen, 24 h exposure to glucose or fructose at 28 mM reduced glucose uptake in L6 cells by 25 to 35%, while l-arabinose did not have an effect (Figure 8c). These findings suggest that l-arabinose, unlike glucose and fructose, does not induce skeletal muscle insulin resistance in the HepG2/L6 coculture model. In the triple coculture with intestinal, liver and skeletal muscle cells, no significant differences in insulin-mediated glucose uptake were observed in L6 cells following a 24 h exposure (at Caco-2 level) to different disaccharides at 14 mM (Figure 8d).

#### 3.3.2. Insulin Sensitivity Determined with Cellular Bioenergetics

In an experiment to test the effects of insulin pre-treatment on glucose-induced ECAR in L6 cells, the physiological insulin concentration of 50 μIU/mL was found to significantly increase the glucose-induced ECAR response by 36%, whereas the glucose-induced response was only 16% and not significantly higher after 1.67 × 10^4^ μIU/mL insulin pre-treatment (Figure 9a,b). High glucose (28 mM) pre-treatment for 24 h did not significantly affect the insulin-stimulated glucose-induced ECAR response, as this response was only lowered in one of the repetitions. Insulin did not impact the glucose-induced OCR response or the ECAR and OCR responses to the mitochondrial stressors oligomycin, FCCP and rotenone/antimycinA (Figure A1).

## 4. Discussion

We investigated how (rare) sugars impact physiological processes in the gut–liver–muscle axis controlling the glycemic index, more specifically, (I) their brush border digestion, (II) their impact on acute and adaptive metabolic responses to the monosaccharides entering the liver and (III) their effect on skeletal muscle insulin sensitivity. Hereby, the research builds further upon the current knowledge on rare sugars and the existing in vitro models for testing cellular impact of nutrients. Currently, there is only a small number of rare sugars for which the metabolic health impact is known, as mentioned in recent reviews [35,36], and this is the first study to investigate metabolic health effects of the rare sugars D-Glc-α1,2-D-Gal, D-Glc-α1,2-D-Rib and D-Glc-α1,3-L-Ara. In addition, this study has used cellular energetics to obtain knowledge on cellular effects of disaccharides, whereas previous research has mainly focused on conventional monosaccharides [25,37]. Lastly, this study introduces a new triple coculture model and new assay combinations to evaluate skeletal insulin muscle sensitivity in vitro, along with suggestions to improve the model further. Using these approaches, we achieved (I) the identification of differential effects of both conventional and rare disaccharides on the energy metabolism, and (II) improved understanding of how specific monosaccharides impact energy metabolism following acute and chronic exposures, demonstrating differences in metabolic flexibility upon chronic exposure to structurally different monosaccharides.

### 4.1. Rare Disaccharides Are More Slowly Digested than Maltose

Respirometry and HPAEC-PAD glucose quantification indicate differences in digestion between disaccharides in Caco-2 cells, reflected by ECAR responses and changes in glucose concentrations over time. The ECAR responses highlight the importance of the glycosidic bond for digestibility, explaining the more delayed digestion of kojibiose and especially trehalose. This is confirmed by the HPAEC-PAD findings, while these analyses also showed that the monosaccharide composition has an impact. Although the decreased glucose levels upon exposure to D-Glc-α1,2-D-Gal, D-Glc-α1,2-D-Rib and D-Glc-α1,3-L-Ara could be related to the fact that only one molecule of glucose is released during their digestion, we have previously demonstrated that cellular ATP production and the change in disaccharide concentration over time is similar for these sugars in comparison to a mannitol control [38]. Therefore, the decrease in glucose concentration during exposure to Glc-α1,2-D-Gal, D-Glc-α1,2-D-Rib and D-Glc-α1,3-L-Ara is most likely a result of both a lower glucose content of the sugar and delayed digestion. Our findings for maltose, kojibiose and trehalose support the ranking of these glucobioses based on their digestibility with rat intestinal extract [39]. These differences may translate to a reduced glycemic effect of kojibiose and trehalose, relevant in the diabetes context. However, inducibility of brush border enzymes may impact digestion, and it is not yet known whether the tested rare sugars can induce expression of brush border enzymes as described for maltose at 12.5 mM [40]. Diet composition with different macronutrients, fibers and bioactives may impact glycemic responses to specific sugars in vivo, which is an aspect that is not included in this study [41,42,43]. Furthermore, inter-personal differences in brush border expression impact glycemic responses in vivo, and are highly relevant for trehalose digestion as intestinal trehalase expression differs considerably between individuals [44]. As a consequence, blood glucose responses to trehalose are small in individuals with low trehalase activity and significantly higher in individuals with high trehalase activity [44].

### 4.2. Different Sugars Influence Short-Term Aerobic and Anaerobic Hepatic Metabolism

Differences in disaccharide digestion result in a different flow of monosaccharides to the liver. Our results suggest that these monosaccharides provide different acute effects on cellular metabolism in the liver, with increased ATP production from glucose and fructose mediated via anaerobic glycolysis. These findings can be linked to studies reporting that glucose and fructose contribute to lactate production, which is less the case with galactose [45,46,47,48]. The absence of an acute sugar-induced increase in mitochondrial ATP production may be explained by the cancerous nature of the cells with a tendency to be more dependent on glycolysis [49], access to alternative (non-sugar) substrates that are depleted after longer exposures and in the case of galactose, an inefficient galactose metabolism in HepG2 cells [48]. Sugars with a larger acute impact on liver metabolism such as glucose and fructose may not be problematic in moderate amounts, but should not exceed amounts that can be metabolized for energy production, as excessive hepatic acetyl-CoA beyond energy may be used for synthesis of cholesterol and triglycerides with adverse influences on metabolic health [50]. This has been observed in interventions with high concentrations of fructose [51], and upon skeletal muscle insulin resistance when accumulation of blood glucose increases the flow of glucose to the liver [16]. It should be mentioned that ATP production measured with the Seahorse is different from the intracellular ATP content, which can be depleted following high-fructose exposures, potentially resulting in excessive uric acid production as metabolite contributing to oxidative stress and endothelial dysfunction [52,53]. Measurement for these adverse effects are not included in our setup.

### 4.3. Chronic Replacement of Glucose in the Culture Stage Alters the Hepatic Energy Metabolism

Using different applications of cellular bioenergetics, we observed that chronic exposure to specific monosaccharides alters basal metabolism and sugar-specific responses, indicative of altered capacities to metabolize metabolic substrates. Although more relevant in the context of metabolic diseases than a single acute dose response, chronic exposure is rarely investigated using cellular in vitro models. The few previous studies on chronic exposure to particular monosaccharides showed that monocultures of L6 and HepG2 cells become more responsive to mitochondrial toxicants following chronic pre-treatment with galactose [48,54]. Our observations suggest that chronic monosaccharide exposures also impact other aspects (e.g., anaerobic glycolysis) of the cellular energy metabolism and could be relevant in the metabolic health context.

A major finding from both the respirometry and resazurin experiments is that HepG2 cells always (independent of the monosaccharide pre-treatment) respond significantly to glucose and fructose, though not to l-arabinose. These findings confirm the non-metabolizable nature of l-arabinose [55] and show that metabolic conversion of this sugar cannot be induced by chronic exposure to l-arabinose. However, multiple other responses are altered upon chronic exposures and can be divided in general metabolic adaptations and sugar-specific responses.

General adaptations include the increase in basal aerobic metabolism (cellular bioenergetics) in cells pre-treated with galactose, as well as the higher basal energy production (resazurin assay) in cells chronically exposed to galactose or l-arabinose. These changes suggest that chronic exposure to galactose and l-arabinose skews cells towards a more efficient aerobic metabolism. The adaptation towards a more aerobic metabolism could be the result of an improved mitochondrial function and may either be an effect of the sugars themselves or an effect caused by glucose retraction, thereby forcing cells to use other substrates present in the medium [48] (such as glutamine and other amino acids) or energy stores inside the cells (such as glycogen, fatty acids and proteins). In this context, mitochondrial function is an important indicator for metabolic health that is improved by insulin [22]. Improved mitochondrial function may facilitate fatty acid oxidation and thereby prevent accumulation of intrahepatic and intramuscular fatty acids capable of inducing local insulin resistance [56,57]. In contrast, hyperglycemia impairs mitochondrial function in cardiomyocytes and rabbit liver [58,59]. However, it should be mentioned that cancerous nature of our cell lines may be partly responsible for the relatively inactive mitochondrial metabolism in cells pre-treated with glucose, since cancer cells rely more on anaerobic glycolysis even when oxygen is present [49].

Interesting sugar-specific responses include (I) altered glucose-induced ECAR and OCR response in galactose pre-treated cells (cellular bioenergetics), (II) a large response to galactose upon galactose pre-treatment (resazurin assay) and (III) an increase in energy generated from glucose in cells pre-treated with fructose (resazurin assay).

Chronic exposure to galactose induced a number of changes related to the glucose metabolism, which was visible as a (I) higher glucose-induced ECAR response, (II) a larger glucose-induced decrease in OCR and (III) a larger difference between the acute glucose-induced and fructose-induced ECAR response. Together, these findings suggest that chronic galactose exposure improves the capacity of cells to metabolize glucose, which could be explained by an increase in glucose uptake and mitochondrial capacity, and contributes to improved glucose handling [60]. Improved handling of glucose may prevent hyperglycemia, as illustrated in vivo by rapid glucose clearance in athletes (who generally have a high mitochondrial capacity, large glycogen stores and above average muscle mass) during hyperinsulinemic-euglycemic clamps [61]. Enhanced hepatic glycolysis as we observed following chronic galactose pre-treatment can result in lower blood glucose levels via consumption of glucose and inhibition of hepatic gluconeogenesis via the enzyme fructose-2,6-biphosphate [62]. This finding may also be an indication of improved metabolic flexibility following chronic galactose pre-treatment, considering the enhanced switch from aerobic metabolism to anaerobic glycolysis. Collectively, cellular alterations upon chronic glucose replacement can impact health by improving metabolism of metabolic substrates, most importantly glucose and fatty acids. However, it should be mentioned that substrate availability will not be affected as much in vivo, even upon permanent replacement of dietary sugars, considering glucose present in the circulation and a variety of nutrients provided by the diet.

The alterations of sugar-specific responses observed with the resazurin assay cannot be coupled directly to metabolic health, but can potentially be explained by changes in enzymatic activity. Chronic exposure to galactose resulted in 75% more resazurin conversion than chronic exposure to glucose when extra galactose is added to the cells, which may, along with a higher basal metabolism, be the result of the upregulation of enzymes involved in the Leloir pathway, as reported in yeast when galactose is present as the only carbohydrate source for an extended period [63]. Davit–Spraul et al. reported that a similar adaptation occurs in HepG2 cells, resulting in enhanced conversion of galactose to glucose as well as an increased activity of galactose-1-phosphate-uridyltransferase (GALT) and glucose-6-phosphate-dehydrogenase (G6PDH) as important enzymes within galactose metabolism [64]. Likewise, the increased glucose response following chronic fructose exposure—despite the initially similar basal energy levels—could be explained by the stimulation of hepatic glucokinase, as it has been reported that this enzyme is stimulated even by small amounts of fructose/fructose-1-phosphate, resulting in an increased formation of glycolytic intermediates [65].

### 4.4. Sugars Differentially Impact Skeletal Muscle Insulin Sensitivity?

Insulin treatment reduces glucose concentrations in the cell medium, and hence enhances glucose uptake in muscle cells, which is in line with the well-described cellular role of insulin [66]. Cellular respirometry with L6 cells confirms that insulin successfully increased cellular uptake and metabolic conversion of glucose, as insulin pre-treatment increased the glucose-stimulated ECAR response in L6 cells. Whereas insulin was effective in stimulating glucose uptake, pre-treatment with 25 mM glucose resulted in a reduced glucose uptake of ±25% in the presence of insulin compared to cells pre-treated with 5.5 mM glucose, suggesting that high glucose exposure was able to trigger insulin resistance, in line with previous publications using L6 muscle cells or adipocytes [67,68]. Although these monoculture effects on glucose uptake and the insulin signaling pathway are a confirmation of previous knowledge, we also studied the impact of different sugars on insulin sensitivity using a coculture model of intestinal, hepatic and skeletal muscle cells.

Glucose and fructose exposure at the HepG2 level in a HepG2/L6 coculture had a similar inhibitory effect on glucose uptake in L6 cells despite the indirect exposure, showing that the effect is not specific for glucose and that the hepatic metabolism may contribute to peripheral insulin resistance as well. It is unclear which mechanism would mediate skeletal muscle insulin resistance in our setup, but it is known that interruption of the hepatic parasympathetic reflex (preventing release of hepatic insulin-sensitizing substance) and antagonism of hepatic nitric oxide synthase can both cause skeletal muscle insulin resistance [69]. In addition, inflammatory cytokines and saturated fatty acids can contribute insulin resistance in skeletal muscle [70], and may be produced by the liver upon high sugar exposures [71]. The absence of an insulin resistance response to l-arabinose exposure suggests that this sugar and sugars containing l-arabinose may be healthier alternatives.

In the triple coculture, however, an effect of disaccharides (including maltose) on insulin-mediated glucose uptake was not visible, which may be explained by higher background concentrations of glucose in this setup. This may mask effects in the triple coculture models compared to the other models, since disaccharides provide more subtle effects due to the delayed release of monosaccharides. Furthermore, the absence of an effect could be due to the intestinal layer reducing the impact of the sugars on the skeletal muscle cells, for example, by metabolizing part of the monosaccharides and thereby reducing the impact of sugars on cocultured liver and skeletal muscle cells. In this latter scenario, the in vivo impact of all the tested disaccharides may be limited as well. Other experimental setups in which longer and/or repeated exposures are applied in media with a small background of glucose may then be required to detect relatively subtle differences between disaccharides. It is important to note that the physiology of Caco-2, HepG2 and L6 cells was not affected by coculture, which potentially allows those extended exposures.

In this context, it is important to realize that glucose homeostasis in vivo is impacted by the intestinal microbiota as well [72]. It has been shown that certain rare sugars, such as D-Glc-α1,3-L-Ara and kojibiose can have prebiotic effects [73], thereby further enlarging their potential as sugar replacers with a more beneficial impact on glucose homeostasis. In contrast, trehalose as one of the more beneficial sugars regarding delayed glucose release in our study was previously found to promote growth of pathogenic gut bacteria and adversely impact alpha diversity of the gut microbiota [73,74]. The microbial aspects should be part of the final evaluation of the health potential of a specific rare sugar. Likewise, other safety aspects have to be taken into account. Rare sugars such as trehalose can impair cellular uptake of conventional sugars [75], which theoretically could both be beneficial in decreasing their adverse metabolic impact and be problematic by reducing glucose availability for glucose-dependent tissues (such as erythrocytes). Currently, there is mostly scientific evidence for the beneficial impact of reducing cellular uptake of conventional sugars such as the glucose-lowering effect of allulose and the anti-cancer effect of allose via interference with the tumor glucose metabolism [76,77].

### 4.5. Rare Sugars with the Largest Health Potential

Our findings on disaccharide digestion, energy metabolism and insulin sensitivity have contributed to some insights in the health perspectives of rare sugars. Firstly, l-arabinose showed no adverse impact on energy metabolism or insulin sensitivity in our models, and although they are not per se rare in nature, we demonstrated that a nigerose analogue with l-arabinose also lacks the adverse metabolic effects associated with conventional sugars. Furthermore, we have shown that multiple rare disaccharides display considerably delayed digestion rates, amongst which D-Glc-α1,2-D-Rib and D-Glc-α1,3-L-Ara may be especially promising because of their delayed digestion and unconventional monosaccharide composition. Nevertheless, the full potential of these sugars remains to be determined in in vitro insulin sensitivity experiments with repeated exposures and finally in vivo.

### 4.6. Model Suitability and Future Perspectives

Coculture models were used to mimic inter-organ crosstalk and the complexity of metabolic health. Although our cell models cannot simulate whole-body metabolism, the intestine, liver and skeletal muscle are arguably the most important tissues for sugar metabolism and digestion, as well as the organs necessary to simulate hepatic and peripheral insulin resistance as the primary diagnostic criteria for diabetes [6,78,79]. The addition of adipocytes, immune cells, pancreatic β-cells and glucose-dependent tissues (e.g., erythrocytes) may strengthen the model further. Furthermore, chronic (as performed in HepG2 monocultures) and repeated sugar exposures in coculture models may provide additional insight in metabolic health.

L6 myotubes of rat origin are a frequently used model to study insulin sensitivity [80,81]. Although rats differ from humans, Wistar rats can develop insulin resistance following exposures that impact insulin sensitivity in humans [82]. Although L6 myotubes have increased expressions of glucose transporters compared to primary human myocytes [83], insulin resistance in skeletal muscle cells is more closely related to alterations in GLUT4 translocation than GLUT4 expression [67]. Nevertheless, comparative studies reported that L6 rat myotubes also experience a larger insulin-mediated effect on glucose uptake than primary human skeletal muscle cells [83,84]. However, GLUT4 expression necessary for the pronounced insulin-dependent glucose uptake in L6 myotubes is obtained during differentiation of the cells [85], whereas our cells were likely not fully differentiated. Our L6 cells have been exposed under serum-free conditions, but were not pre-treated with standardized low serum conditions that induce differentiation [85]. Therefore, our L6 were still in a myoblast stage and this model will underestimate rather than overestimate the insulin-dependent glucose uptake in primary human skeletal muscle cells, as confirmed by our finding that insulin increased glucose uptake only two-fold. This reduced window of effect may partly explain why no effects of disaccharides on insulin sensitivity were observed in the triple coculture model, and highlights the importance of future research with models having a confirmed potent GLUT4 expression.

Final experiments were performed under physiological conditions which include culturing (I) HepG2 and L6 cells on 5.5 mM glucose as normal glucose concentration in the bloodstream [86]; (II) using insulin concentrations within the normal range for fasted (±8.3 μIU/mL) and stimulated (67 μIU/mL after an oral glucose challenge) conditions [87]; and (III) performing intestinal exposures with disaccharides at 14 mM, which is well below luminal glucose peak concentration observed after a meal [88]. In contrast, experiments building up to the triple coculture involving direct exposures to L6 and HepG2 cells were performed with a sugar concentration (28 mM) that exceeds portal vein glucose concentrations found in feeding trials (8 mM) [89]. Moreover, this concentration is higher than the cut-off value for the diagnosis of diabetes determined 2 h after an oral glucose challenge (11.1 mM) [86], which may explain the rapid development of metabolic complications in these cell models. These high monosaccharide exposures were used to test if metabolic complications could be induced in a worst-case scenario and serve as preparations for the disaccharide comparison in coculture models with an intestinal compartment.

## 5. Conclusions

Insulin resistance in skeletal muscles contributes to type 2 diabetes development by interfering with peripheral glucose uptake, although it is not completely understood whether and how different sugars impact insulin resistance. Our results demonstrate that glucose and fructose, unlike l-arabinose, are metabolized efficiently in an anaerobic manner in HepG2 cells and are capable of interfering with skeletal muscle insulin resistance when exposed for 24 h at high concentrations in a muscle or liver/muscle model. The rare disaccharides trehalose, D-Glc-α1,2-D-Gal, D-Glc-α1,2-D-Rib and D-Glc-α1,3-L-Ara are slowly digested in comparison to maltose, and the glucose released upon their digestion is unlikely to adversely impact insulin sensitivity. However, the direct link between intestinal disaccharide digestion rate and insulin sensitivity could not be made as none of the tested disaccharides induced skeletal muscle insulin resistance in the gut/liver/muscle model. Therefore, repeated exposures in realistic in vitro models and finally in vivo experiments are needed to further investigate this hypothesis. The most promising rare sugars would need to be subjected to well-designed clinical trials, and their integration in the diet needs to take both multi-endpoint health aspects among different target groups of consumers, as well as food production aspects, into account.

## Figures and Tables

**Figure 1 nutrients-15-01593-f001:**
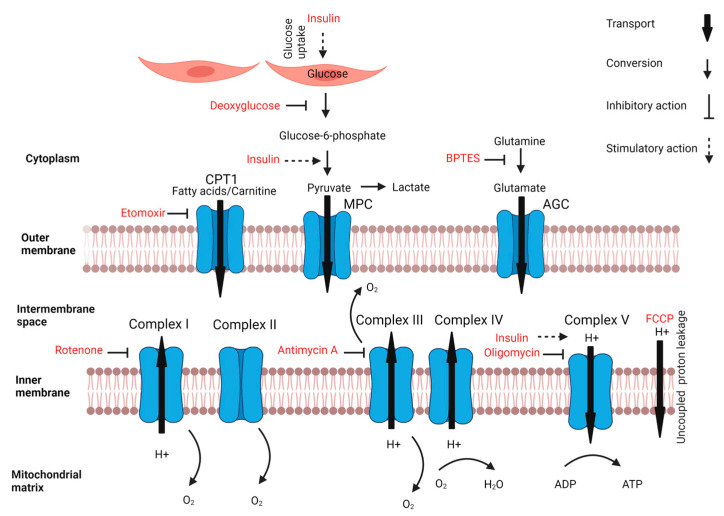
Effects of insulin and metabolic stressors on mitochondrial and macronutrient metabolism (Biorender).

**Figure 2 nutrients-15-01593-f002:**
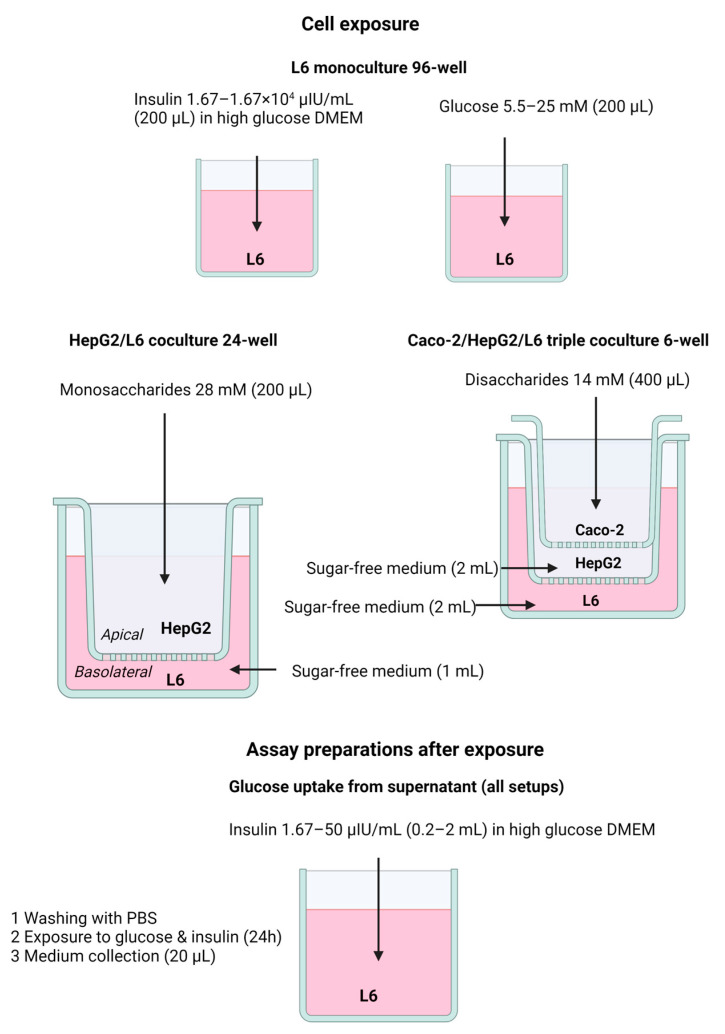
Schematic overview showing the different exposure conditions in the monoculture, coculture and triple coculture setup, as well as the preparation steps taken for the different methods to determine insulin sensitivity.

**Figure 3 nutrients-15-01593-f003:**
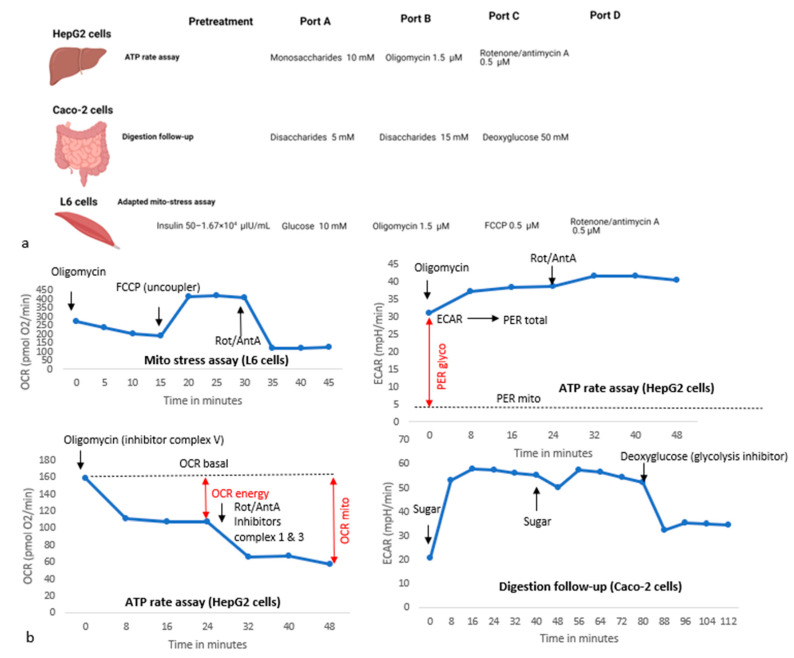
**Overview figure for exposures and setups in cellular bioenergetics experiments.** Subfigures show the stimuli and stressors that were used in different setups (**a**) as well examples of general responses to different stressors (**b**), based on our experimental data. Measures for calculation of ATP rates are indicated in the graphs for the ATP rate assay.

**Figure 4 nutrients-15-01593-f004:**
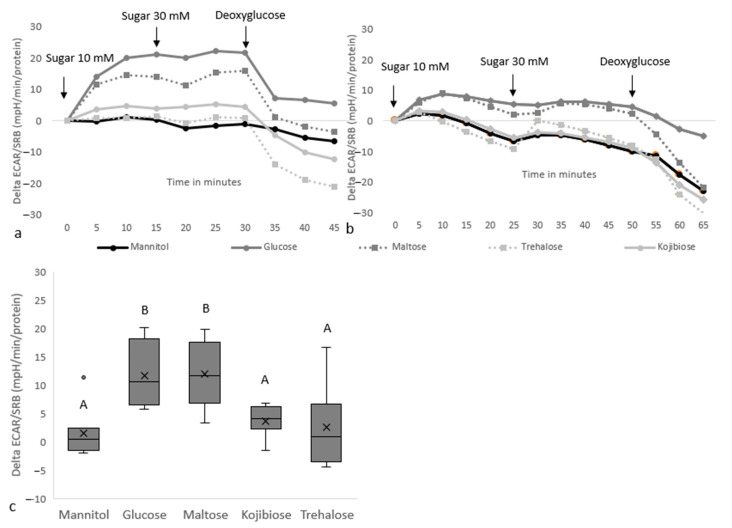
**The effect of disaccharide injections on protein-corrected ECAR responses in differentiated Caco-2 cells.** Subfigures (**a**,**b**) show the ECAR profiles of individual experiments with 3–5 replicates per condition, whereas (**c**) shows the boxplots for the 10 min ECAR increase upon injection of 10 mM sugar and consists of the pooled data from graphs (**a**,**b**). The assay medium at the start of the assay consisted of XF base medium, without sugars, L-glutamine or pyruvate. B indicates a statistically significant (*p* < 0.05) difference compared to the mannitol control, whereas A indicates that there is no significant difference compared to mannitol.

**Figure 5 nutrients-15-01593-f005:**
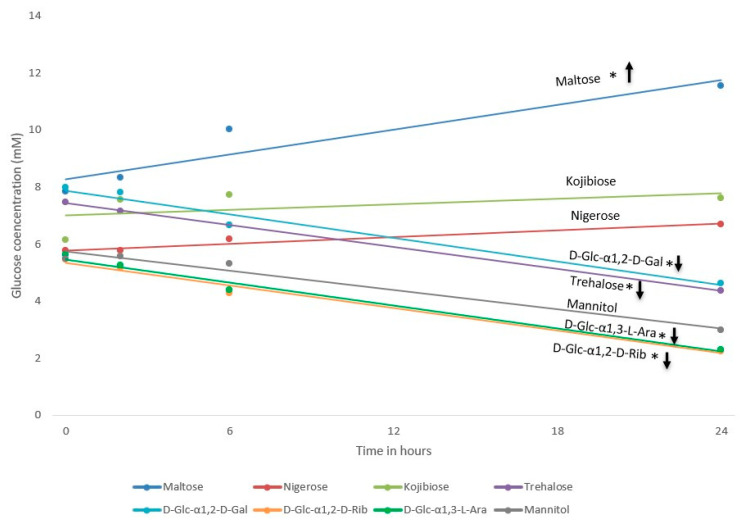
Changes in glucose concentrations within 24 h, following disaccharide exposures in a Caco-2/HepG2/L6 triple coculture model. This figure visualizes the glucose concentration in the Caco-2 medium, sampled at 0, 2, 4 and 24 h exposure. Data were generated from four different wells (each from a separate plate) per condition with * indicating a statistically significant (*p* < 0.05) change in glucose concentration within 24 h. Arrows indicate the direction (increase versus decrease) of the significant changes in glucose concentration.

**Figure 6 nutrients-15-01593-f006:**
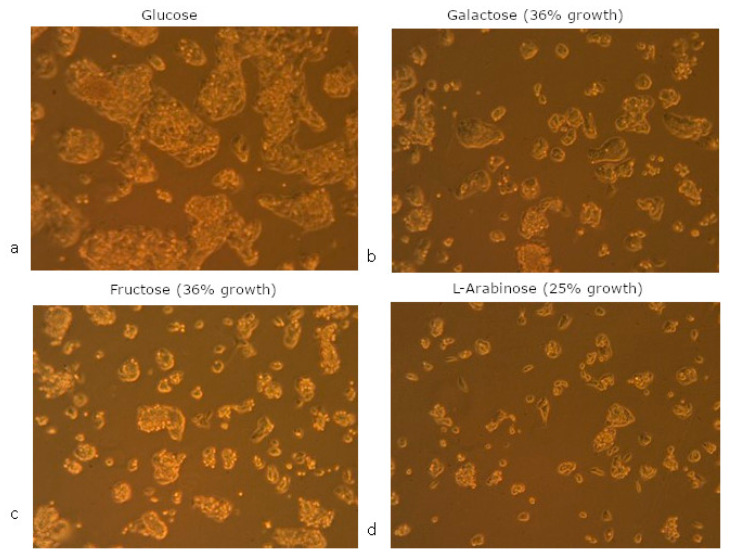
Morphology of HepG2 cells following chronic exposure to 5.5 mM glucose (**a**), galactose (**b**), fructose (**c**) or l-arabinose (**d**). Microscopic pictures (10× magnification) were taken after 2 weeks of culture in the different media and 2 days after splitting. Cell growth on alternative media is indicated as % compared to glucose, based on cell count and time until ±80% confluency. Chronic exposures were performed in sugar-free medium with FBS and L-glutamine.

**Figure 7 nutrients-15-01593-f007:**
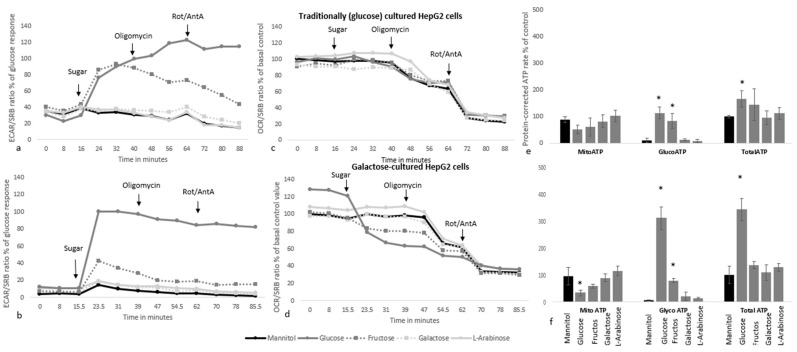
The acute response to different sugars at 10 mM in HepG2 cells cultured under standard conditions (5.5 mM glucose, (**a**,**c**,**e**)) and after chronic galactose (5.5 mM, (**b**,**d**,**f**)) pre-treatment. Subfigures show the protein-corrected effects on the ECAR profile (**a**,**b**), the OCR profile normalized to the basal level in control cells (**c**,**d**) and ATP rates as % of total ATP rate during mannitol exposure (**e**,**f**). The assay medium at the start of the assay consisted of XF base medium, without sugars, L-glutamine or pyruvate. Data were generated from a total of eight wells per condition spread over two plates (**a**,**c**,**e**), or from four wells of a single plate (**b**,**d**,**f**)). * indicates statistical significance (*p* < 0.05) compared to the mannitol response.

**Figure 8 nutrients-15-01593-f008:**
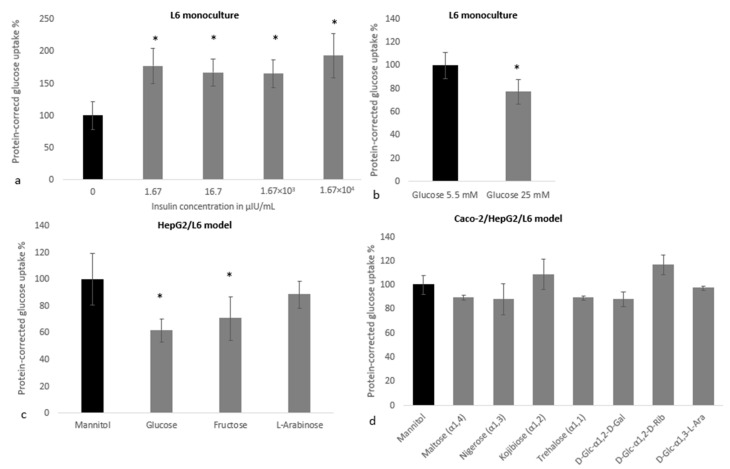
The effect of sugars and insulin on insulin-mediated glucose uptake in L6 cells determined with the GOD-POD assay, using models with different complexity. (**a**) Shows the effect of insulin on glucose uptake from an experiment with six replicates from a single plate. The other subfigures show how 24 h pre-treatment with monosaccharides (28 mM) or disaccharides (14 mM) impact insulin-mediated (50 μIU/mL) glucose uptake in a L6 monoculture model ((**b**): three independent plates for a total of nine replicates), HepG2/L6 coculture model ((**c**): two independent plates for a total of eight replicates) and Caco-2/HepG2/L6 triple coculture model ((**d**): at least three replicates per condition). Data were corrected for SRB (**a**–**c**) or average protein concentration within conditions measured with the Bio-Rad DC protein assay (**d**), and are presented as mean ± standard deviation with * indicating significant differences (*p* < 0.05) compared to the mannitol control.

**Figure 9 nutrients-15-01593-f009:**
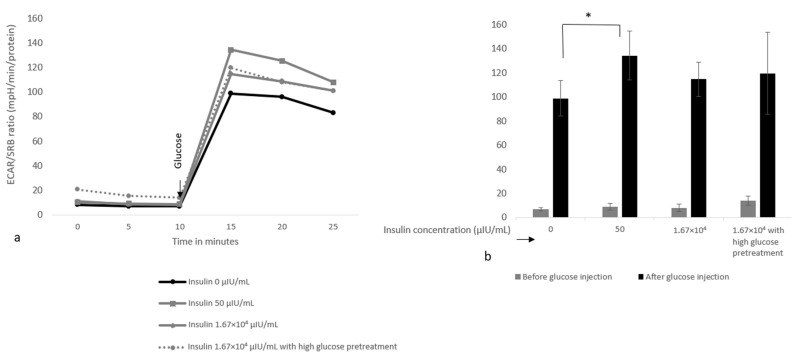
The effect of insulin concentrations and high glucose pre-treatment on the glucose-induced ECAR response in L6 cells. Subfigures show the protein-corrected glucose-induced ECAR response as a Seahorse profile (**a**) and a bar graph. (**b**) The assay medium at the start of the assay consisted of XF base medium, without sugars, glutamine or pyruvate. (**b**) Shows bars for the 10 min (gray before bar) and 15 (black after bar) minute timepoint. Data were generated from 2 independent plates for a total of 10 wells per condition, and are presented as mean ± standard deviation with * indicating significant differences (*p* < 0.05).

**Table 1 nutrients-15-01593-t001:** Protein-corrected resazurin conversion in HepG2 cells cultured in media with 5.5 mM monosaccharides (glucose, fructose, galactose or l-arabinose) following 24 h exposure to these different sugars, presented as % compared to mannitol exposure in glucose pre-treated HepG2 cells. Data were generated from 3 experiments with a total of 18 replicates and are presented as mean ± standard deviation with * indicating a statistically significant (*p* < 0.05) effect of the monosaccharide exposure compared to mannitol and # indicating an effect of the pre-treatment compared to glucose.

Chronic Pre-Treatment (5.5 mM)				
	Glucose	Fructose	Galactose	l-arabinose
24 h Exposures	resazurin/SRB ratio %			
Mannitol	100 ± 15	118 ± 32	146 ± 28 #	137 ± 42 #
Glucose	158 ± 24 *	204 ± 39 * #	204 ± 25 * #	195 ± 40 * #
Fructose	150 ± 39 *	171 ± 32 *	184 ± 43 * #	207 ± 45 * #
Galactose	105 ± 16	128 ± 34	180 ± 43 #	150 ± 26 #
l-arabinose	101 ± 19	117 ± 26	124 ± 27 #	113 ± 26

## Data Availability

Data is included in the article submission to Nutrients. Samples generated from our cell lines are registered in a database: ‘Biobank Vakgroep Levensmiddelentechnology, Voedselveiligheid en Gezondheid’ with accesscode BB190156.

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
