# Peer review of "Rare Sugar Metabolism and Impact on Insulin Sensitivity along the Gut–Liver–Muscle Axis In Vitro"

_nutrients, 2023, doi:10.3390/nu15071593_

Round 1
Reviewer 1 Report
In study by van Laar et al. to investigate the impact of rare sugars on metabolic health, especially skeletal muscle insulin sensitivity, acute sugar-induced changes in cellular bioenergetics and metabolic adaptation of liver cells on monosaccharides were studied. The experimental setting of the manuscript is interesting.
The manuscript's structure is correct and typical for this type of study.
Introduction
The background of the study is sufficiently described and clearly states the aim of the study.
Point 1.
Page 2, line 85-96 -> Please, consider moving to subsequent sections. It is not clear why elements of the methodology and preliminary results are included in the Introduction.
Material and methods.
Material and methods are described in detail, and filled with appropriate Figures.
Results
The restults are synthetically but sufficiently and understandably presented
Discusion and Conclusion.
The discussion is well written and based on other relative studies. However, the authors should more highlight the novelty of the present study and distinction from previous studies.
Reviewer 2 Report
Brief Summary
In the study by Laar et al., the authors compared the effects of rare and conventional sugars on metabolic health in several dimensions, including speed of digestion, sugar-induced changes in cellular bioenergetics and metabolic adaptation, and skeletal muscle insulin sensitivity with a conclusion that rare sugars are likely more metabolically beneficial compared to conventional sugars.
The study investigated a novel topic of rare sugar metabolic impact, which could provide an important foundation for further investigations.
General Concept Comments
1. The authors should also discuss the potential negative effects of rare sugars on the function of obligate glucose users (e.g., CNS, kidney medulla, red blood cells, etc.) to facilitate a more balanced view on this topic.
2. Do rare sugars have any positive or negative effects on microbiota health and diversity? They could play an important role in metabolic health in vivo as well.
Specific Comments
None
Reviewer 3 Report
This is an interesting article
Introduction is comprehensive.
Materials and Methods are clear and well written.
Results are properly explained.
Discussion is appropriate.
References need to be significantly improved. There are only 3 published studies from years 2021-2023 listed in the references. This needs to be more updated. Please include the following articles below.
- Regarding the role of fructose:
The Role of Fructose as a Cardiovascular Risk Factor: An Update. Metabolites. 2022 Jan 12;12(1):67. doi: 10.3390/metabo12010067.
- Regarding a modern approach to glucose metbolism disorders:
Glucose Metabolism Disorders: Challenges and Opportunities for Diagnosis and Treatment. Metabolites. 2022 Jul 29;12(8):712. doi: 10.3390/metabo12080712.
- Regarding the role of Skeletal Muscle in Insulin Resistance:
Exploring the Role of Skeletal Muscle in Insulin Resistance: Lessons from Cultured Cells to Animal Models. Int J Mol Sci. 2021 Aug 28;22(17):9327. doi: 10.3390/ijms22179327.
- Regarding a modern approach to phenotype subjects with insulin resistance at high risk for type-2 diabetes:
New Sub-Phenotyping of Subjects at High Risk of Type 2 Diabetes: What Are the Potential Clinical Implications? Diabetes Ther. 2021 Jun;12(6):1605-1611. doi: 10.1007/s13300-021-01065-3.
